

# Improving temporal smoothness and snapshot quality in dynamic network community discovery using NOME algorithm

Lei Cai[1], Jincheng Zhou[2] and Dan Wang[3]

[1] State Key Laboratory of Public Big Data, College of Computer Science and Technology, Guizhou University, Guiyang, China
[2] Key Laboratory of Complex Systems and Intelligent Optimization of Guizhou Province, School of Computer and Information, Qiannan Normal University for Nationalities, Duyun, China
[3] School of Mathematics and Statistics, Qiannan Normal University for Nationalities, Duyun, China

Corresponding author
Jincheng Zhou, zjc81@sgmtu.edu.cn

## ABSTRACT

The goal of dynamic community discovery is to quickly and accurately mine the network structure for individuals with similar attributes for classification. Correct classification can effectively help us screen out more desired results, and it also reveals the laws of dynamic network changes. We propose a dynamic community discovery algorithm, NOME, based on node occupancy assignment and multi-objective evolutionary clustering. NOME adopts the multi-objective evolutionary algorithm MOEA/D framework based on decomposition, which can simultaneously decompose the two objective functions of modularization and normalized mutual information into multiple single-objective problems. In this algorithm, we use a Physarum-based network model to initialize populations, and each population represents a group of community-divided solutions. The evolution of the population uses the crossover and mutation operations of the genome matrix. To make the population in the evolution process closer to a better community division result, we develop a new strategy for node occupancy assignment and cooperate with mutation operators, aiming at the boundary nodes in the connection between the community and the connection between communities, by calculating the comparison node. The occupancy rate of the community with the neighbor node, the node is assigned to the community with the highest occupancy rate, and the authenticity of the community division is improved. In addition, to select high-quality final solutions from candidate solutions, we use a rationalized selection strategy from the external population size to obtain better time costs through smaller snapshot quality loss. Finally, comparative experiments with other representative dynamic community detection algorithms on synthetic and real datasets show that our proposed method has a better balance between snapshot quality and time cost.

# INTRODUCTION

With the diversified development of social platforms, the sharp increase of various social software and social media users, thus generating a huge amount of data, wanting to mine valuable network information from the huge amount of network data, the study of complex networks becomes increasingly important. Community structure, as one of the important properties of complex networks, has received the most attention from researchers (*Zhou, Wu & Jin, 2018*; *Fortunato, 2010*; *Fortunato & Hric, 2016*). Some early researchers treated real networks as static networks; however, in the real world, the growth of network data changes dynamically over time, and the number of nodes and edges increases or decreases accordingly, and thus the structure of the community changes as well. For example, in the real world, some algorithms have been used to identify people with the same interests in social networks (*Tajeuna, Bouguessa & Wang, 2019*), to find the writer's collaboration networks of writers formed by works done by academics in collaborating with other authors (*Girvan & Newman, 2002*), and to predict protein functions (*Lee, Gross & Lee, 2013*) and applied to recommendation systems (*Jun et al., 2019*). Thus dynamic community discovery is gradually becoming a major research direction for researchers in many fields (*Cai et al., 2016*; *Jin et al., 2015*; *Perc et al., 2017*).

Over the last two decades, more and more researchers have started to study dynamic networks, and there are more and more approaches to solving the problem of community partitioning in dynamic networks. *Besharatnia, Talebpour & Aliakbary (2022)* combined the improved gray wolf optimizer algorithm and the label propagation algorithm for better performance. This method introduces a certain randomness which can increase the searchability of the algorithm. However, this can also make the algorithm unpredictable and difficult to analyze theoretically. *Jiang & Zhang (2022)* proposed a dynamic community detection algorithm based on assignment and segmentation to reduce error accumulation in incremental methods and detected the final community structure by merging and optimizing splitting. This method is limited by the accuracy of community division at adjacent moments in the process of community discovery.

To improve the accuracy and stability of community discovery, the method based on the evolutionary clustering algorithm has become one of the mainstream algorithms for solving dynamic network community discovery algorithms. The evolutionary clustering algorithm was proposed by *Chakrabarti, Kumar & Tomkins (2006)* for revealing the continuous change of the network over time, called the concept of temporal smoothness of network evolution, who considered the evolutionary clustering method influenced by historical network structure and historical community structure information. Later *Folino & Pizzuti (2014)*. proposed a community detection method based on evolutionary clustering and considered the detection of time-smoothed communities as a multi-objective problem. The basic idea is to ensure that the community quality of the network structure divided at the current moment is as high as possible at the time of community discovery, while the network structure divided at two adjacent time nodes does not change excessively. Their work demonstrated the effectiveness and accuracy of multi-objective optimization algorithms in solving the problem of community discovery in dynamic networks, and

as a result, many methods based on multi-objective evolutionary algorithms have been designed to detect communities in dynamic networks.

In multi-objective optimization-based dynamic network community discovery, most researchers use modularity (Q) (*Newman & Girvan, 2004*) based on the quality of network community segmentation at the current time step and normalized mutual information (NMI) (*Danon et al., 2005*) based on two consecutive time steps as two conflicting objective functions in the algorithm to evaluate the accuracy and reliability of dynamic network community segmentation. *Folino & Pizzuti (2010)* proposed a dynamic multi-objective genetic algorithm (DYN-MOGA) based on non-dominated ranking (*Deb et al., 2002*) (NSGA-II) that uses modularity and NMI as optimization objectives to solve the community detection problem in dynamic networks. It automatically provides a solution representing the best trade-off between the obtained clustering accuracy and the deviation from one time step to successive steps. *Niu, Si & Wu (2017)* used a label-based dynamic multi-objective genetic algorithm to detect community structures in dynamic networks with two objectives maximizing snapshot quality and minimizing time cost. *Wang et al. (2019)* assessed the accuracy of community discovery by proposing a population intelligence approach based on an evolutionary clustering framework and labeling using a discrete particle swarm algorithm that incorporates label propagation and genetic algorithm improvements, while also introducing modularity and normalized mutual information. This method can better identify the quality of the community structure, but the calculation speed needs to be improved. To reduce the computational complexity of MOEAs, *Zhang & Li (2007)* proposed a decomposition framework MOEA/D for multi-objective evolutionary algorithms, which decomposes the multi-objective optimization problem into multiple scalar optimization sub-problems and optimizes these sub-problems simultaneously. *Ma et al. (2014)* proposed a decomposition-based dynamic social network (DYNDMLS) using the MOEA/D framework for the first time, and also using modularity, and NMI as optimization goals for dynamic community discovery. On this basis, *Gao et al. (2018a)* and *Gao et al. (2018b)* proposed a decomposition-based multi-objective discrete particle swarm optimization algorithm to discover dynamic structures, combining the particle swarm optimization algorithm with the MOEA/D framework to optimize both modularity density and NMI. However, it has the disadvantage of undesired premature shrinkage and grain monotonicity due to highly selective stress. *Wang, Song & Sun (2022)* proposed a dynamic community detection algorithm based on a multi-objective selectable path-guided pity beetle algorithm in order to improve modularity and subsequent NMI at each time step by combining the MOEA/D framework with an improved identification method for module density and an individual implementation update strategy for neighborhood vector competition. It can minimize the impact of community partitioning at the first time step on subsequent time steps.

The researchers mentioned above have used many evolutionary clustering-based methods to reveal the community structure in dynamic networks. However, they still have some shortcomings in the efficiency of the search for the optimal solution. On the one hand, their operator implements the generation of new candidate solutions but does not avoid the existence of inter-community connectivity between nodes and their most

occupied neighboring communities. On the other hand, the selection strategy based on the optimal solution ignores the better time step partitioned communities by pursuing only the highest quality communities, which leads to a limited search space for subsequent time-steps. To address these issues, under the evolutionary community detection method in dynamic social networks proposed by *Liu et al. (2019)*, we propose a new multi-objective evolutionary algorithm, NOME, which is mainly used to better capture the evolutionary patterns of communities in dynamic networks. The contributions of this article can be summarized as follows: (i) In order to obtain a better community structure, we propose a method of boundary node occupancy assignment cooperating with the mutation operator in the classical genetic algorithm, while employing the genome (*Li, Gao & Pu, 2014*) to represent the network and dividing the community to which the node belongs by calculating the percentage of the node's neighbors in the community. (ii) In order to be able to find a relatively better final solution from the optimal solution, we propose a rationalized selection scheme based on the final Pareto optimal solution set at the current moment to obtain a better temporal smoothness by losing a smaller snapshot quality.

The rest of this article proceeds as follows: Section 2 describes the multi-objective optimization problem, the evaluation function, and the encoding approach for community discovery in dynamic networks. A description of our proposed method is given in Section 3. Then the results of NOME and other advanced algorithms on different datasets are compared in Section 4. Finally, Section 5 summarizes the conclusions.

# BACKGROUND

## Dynamic network problem description

A dynamic network can be defined as a set of multiple static network snapshots $G = \{G^1, G^2, \ldots, G^T\}.G^t = \{V^t, E^t\}$, where $V^t$ is the set of nodes of $G^t.E^t$ is the set of edges of $G^t$, denoting the set of edges at moment $t$ ($t \in [1, T]$) consisting of two different nodes at that moment, and all the nodes and their connected edges form this network structure.

## Dynamic network community discovery problem description

Dynamic network community detection is to find the community structure division of the network structure at each moment, it involves two main objectives in the process of finding, one is to ensure the quality of the community division in the current time step, and the other is to ensure that the community structure changes slowly in the continuous time, *i.e.,* there is no huge change. Suppose that after dynamic network community detection, a network snapshot $G^t$ of the community structure division is obtained at moment $t$, denoted as $C = \{C_1, C_2, \ldots, C_k\}$, where $C_i^t$ denotes the community structure of the $i$- th community division, $i \in [1, k]$. In general, to better accomplish these two objectives, we use the Normalized Mutual Information (NMI) function to detect the similarity of two neighboring temporal community divisions and the modularity function $Q$ to assess the quality of the current community division.

## Normalized mutual information function (NMI)

NMI (*Danon et al., 2005*) is used to detect the similarity of community division between the current time sequence and the previous time sequence, and its formula is shown in Eq. (1).

$$NMI\left(C^t, C^{t-1}\right) = \frac{-2\sum_{i=1}^{m^t}\sum_{j=1}^{m^{t-1}} L_{ij}\log\left(\frac{L_{ij}N}{L_{i.}L_{.j}}\right)}{\sum_{i=1}^{m^t} L_{i.}\log\left(\frac{L_{i.}}{N^t}\right) + \sum_{j=1}^{m^{t-1}} L_{.j}\log\left(\frac{L_{.j}}{N^{t-1}}\right)} \tag{1}$$

where $C^t = \{C_1^t, C_2^t, ..., C_k^t\}$ $C^{t-1} = \{C_1^{t-1}, C_2^{t-1}, ..., C_k^{t-1}\}$ denotes the community structure divided at the moment $t$ ($t$-1) and $L_{ij}$ denotes the number of nodes that are both in community $C_i^t$ and in community $C_j^{t-1}$. $N$ is the number of nodes corresponding to $C^t$, $m^t$ and $m^{t-1}$ denote the number of community divisions in $C_i^t$ and $C_j^{t-1}$. $L_{i.}$ and $L_{.j}$ are the sum of elements in row $i$ and column $j$ of the confusion matrix $L$, respectively, $NMI(C^t, C^{t-1}) = 1$. If $NMI(C^t, C^{t-1}) = 1$, it means that $C^t$ and $C^{t-1}$ have exactly the same community division result; conversely, 0 means that these two communities have completely different division results, so the closer the value of NMI is to 1 the more similar $C^t$ and $C^{t-1}$ are on the surface.

## Modularity function (Q)

The modularity function Q (*Newman & Girvan, 2004*) was proposed by Newman et al. to measure the quality of the division of the community structure, and a larger value of modularity indicates that the result is closer to the real community structure. It is defined as the expected value of the difference between the ratio of the number of edges inside the network community to the number of edges in the whole network and the ratio of the number of arbitrarily connected edges between nodes to the number of edges in the whole network under the same community structure, and its formula is shown in Eq. (2).

$$Q = \frac{1}{2m}\sum_{i=1}^{n}\sum_{j=1}^{n}\left[A_{ij} - \frac{k_i k_j}{2m}\right]\delta\left(c_i, c_j\right) \tag{2}$$

where $m$ denotes the number of edges, $n$ is the number of network vertices, $A_{ij}$ is the adjacency matrix that represents the relationship between edges before node $i$ and node $j$, $k_i$ and $k_j$ are the degrees of nodes, and $c_i$ is the community label of node $i$. If $\delta(c_i, c_j) = 1$, node $i$ and node $j$ are in the same community, otherwise $\delta(c_i, c_j) = 0$. The value range of module degree Q is $[-0.5, 1]$.

## Multi-objective optimization

The multi-objective optimization algorithm can optimize two or more conflicting objectives at the same time. We define the multi-objective problem of community partitioning at time $t$ as

$$\begin{cases} minF\left(C^t\right) = \left[f_1\left(C^t\right), f_2\left(C^t\right), ..., f_m\left(C^t\right)\right] \\ s.t.\, C^t \in \Omega \end{cases} \tag{3}$$

where $f_i(C^t)$ is the i-th ($i = 1, 2, ..., m$) objective function, $m$ is the number of objective functions, $C^t$ is the community structure at time step $t$, and $\Omega$ is the feasible domain of the set of all community structures at time step $t$.

Suppose $C_1^t$ and $C_2^t$ are two feasible solutions to this dynamic network optimization problem. We define the dominance relation of the solution of community partitioning at time $t$ as Eq. (4), and $C_1^t$ is said to dominate $C_2^t$ when and only if all the conditions of formula (4) are satisfied.

$$\begin{cases} \forall i = 1, 2, \ldots, m; \\ f_i(C_1^t) \leq f_i(C_2^t) \wedge \exists j \in [1, m]; \\ f_j(C_1^t) < f_j(C_2^t) \end{cases} \tag{4}$$

If a solution is not dominated by any other solution, the solution is called an non-dominated solution. All non-dominated solutions form a Pareto optimal set (PS), and the set of PS mapped to the target space is called the Pareto front (PF).

## Coding method

In performing the community discovery process, the community of each individual is divided using coding. The main current encoding methods are the string encoding method and the bit-adjacent encoding method (*Gong et al., 2014*). In this article, a combination of string encoding and genomic matrix encoding is used for each individual.

Genome matrix coding refers to the representation of information about the community and node-to-node information in the form of a genome. The individuals for which the community finds a solution to the problem are encoded as $n \times n$ matrix genomes of order $n$, where $n$ equals the number of nodes $|v|$. $e_{ij}$ is used in this article to represent the relationship between node $v_i$ and node $v_j$, while the value of $e_{ij}$ proves whether $v_i$ and $v_j$ belong to an edge within a community or between communities. where $e_{ij} = -1$, indicating that $v_i$ and $v_j$ are connected to each other; $e_{ij} = 0$, indicating that there is no connection between the two nodes; and $e_{ij} = 1$, indicating that $v_i$ and $v_j$ are nodes belonging to the same community.

Figure 1A shows an original network and the original community, which can be encoded into the form of a genome matrix using the genome matrix according to the connection relationship between the network nodes, and then the genes in the matrix were updated after the reorganization (crossover, mutation, and BNO assignment) operation, and finally, the structure of the divided community was obtained by decoding. Figure 1B shows the divided network, and nodes of the same color indicate that they are in the same community. In the reconstructed matrix how to decode as string encoding method, we divide the sub-matrix genes into the same community based on the relationship of inner connection in the matrix until all the inner connection genes that are divided. As the yellow and green areas in the matrix before decoding are the two largest inner-connected sub-matrices we find, and also satisfy all inner-connected genes are divided, so the decoding operation is completed by converting all the divided sub-matrices to string encoding.

## Proposed algorithm

This section proposes an improved algorithm NOME to discover dynamic community structures based on the work on evolutionary community detection in dynamic social networks proposed by *Liu et al. (2019)*. A decomposition-based MOEA/D algorithm

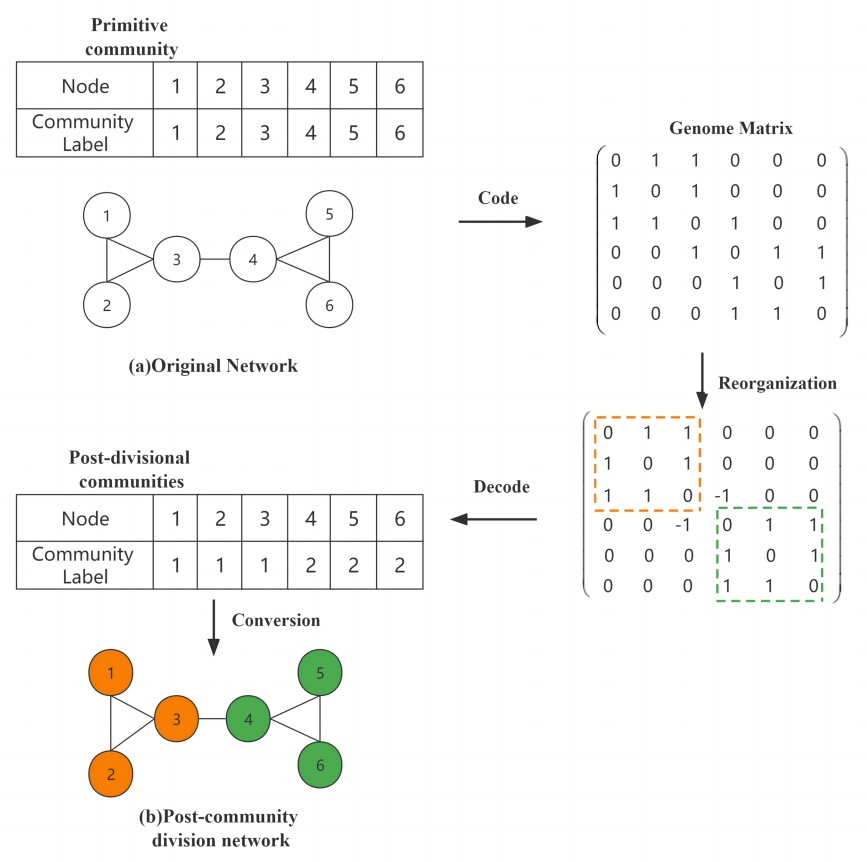

**Figure 1  Community division of the network.**

framework is used to improve the quality of community detection by migrating nodes with respect to their occupancy in neighboring communities. In addition, in order to obtain a better time step for the division of the network structure, NOME uses a rationalized selection scheme to obtain the final solution based on the final Pareto optimal solution set at the current moment. Algorithm 1 shows the overall flow of NOME.

---

**Algorithm 1.** NOME

---

**Input:** dynamic network G $= \{G^1, G^2, \ldots, G^T\}$, population_size, maxgen

**Output:** Community structure for each snapshot $C^t = \{C_1^t, C_2^t, \ldots, C_k^t\}(t=1,2,\ldots,T)$

1 $t = 1$ Single-objective optimization snapshot based on modularity $G^1$ obtained $C^1$

2 **for** t = 2:T **do**

3    // Initialization section

4    Initializing non-dominated solutions $EP = \varnothing$

5    Generate a uniformly distributed weight vector $\lambda = \{\lambda^1, \ldots, \lambda^N\}$

6    Generate neighbor information based on Euclidean distance $B(i) = \{i_1, i_2, \ldots, i_T\}$

7    Initializing the population

8    Initialize reference points $Z* = \{NMI_{\max}, Q_{\max}\}$

9    **for** gen = 1: maxgen **do**

10      // Restructuring component

11      **for** pop_id =1: pop_size **do**

12       Select two different individuals from the neighborhood

13       Perform crossover operations

14       **if** rand(1)<pm **then**

15        Performing mutation operations

16       **else**

17        Execute the boundary node occupancy allocation method

18       **end**

19      Update Neighborhood Information

20      **end**

21      Find non-dominated solutions to add to the EP and remove all dominated solutions from the EP

22      Update reference point z

23    **end**

24 **end**

25 Optimal solutions are obtained using rationalized selection schemes

---

## Boundary node occupancy assignment

To improve the quality of community checking solutions in dynamic networks, we propose the community boundary node occupancy (BNO) assignment method, which can better classify nodes into the correct communities. This operation emphasizes the interconnection of nodes within and between communities, and the BNO is specifically calculated as

$$O_c^i = \begin{cases} \dfrac{B_c^i}{N_c - 1}, i \in c; \\ 0, N_c = 1; \\ \dfrac{B_c^i}{N_c}, others \end{cases} \tag{5}$$

where $O_c^i$ means to calculate the occupancy rate of the current $i$-th node in the neighbor community $c$. $c$ represents the $c$ th community in the node $i$ neighbor community. $B_c^i$ represents the total number of neighbor nodes of node $i$ in the $c$ th neighbor community, and $N_c$ represents the total number of nodes in the $c$ th community. If node $i \in c$, then the node itself needs to be subtracted when calculating the total number of nodes in the $c$-th

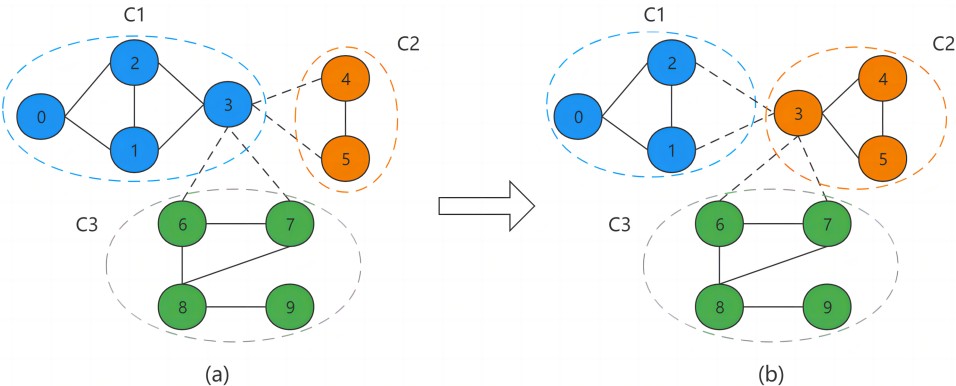

**Figure 2** Boundary node occupancy allocation.

community. If $N_c = 1$, then the occupancy of node $i$ in the $c$-th neighboring community is 0. The value range of $O_c^i$ is between $[0,1]$.

Figure 2 illustrates the process of BNO through an example. Where the intra-community and inter-community connections are represented by solid and dashed lines, respectively, and nodes of the same color indicate being in the same community. Figure 2A shows the structure of the divided network community. Node v3 is a boundary node of community C1. v3′s neighboring nodes are {v1, v2, v4, v5, v6, v7}, where nodes v1 and v2 belong to community C1, v4 and v5 belong to community C2, and v6 and v7 belong to community C3, so its neighboring communities are C2 and C3. For a boundary node, we need to calculate its occupancy in each neighboring community. The calculation by Eq. (5) shows that since node v3 belongs to community C1, it needs to subtract the node itself when calculating the total number of community nodes, so its total number of nodes is 3. And there are two neighbor nodes of v3 in community C1, which are v1 and v2, so the final calculation gets the occupancy of v3 in community C1 as 2/3. Similarly, the occupancy of v3 in community C2 is 1 and in community C3 is 1/2. Finally, through the node occupancy allocation method, node v3 will eventually be assigned to community C2, as in Fig. 2B.

There are three possible cases in the BNO allocation process, and we use different allocation schemes for different cases. The first one is that the occupancy rate of the community to which the boundary node belongs is higher than the occupancy rate of its neighbor community, at which point the node will not be assigned out. The second case is when there are multiple communities with the same maximum occupancy among all neighboring communities of the boundary node, we randomly assign the node to one of them. If one of the communities belongs to the node community, then we will determine whether the node is assigned to the other maximum occupancy community by the probability $Pm$. The third case is when the occupancy rate of the community to which the boundary node belongs is lower than the occupancy rate of its neighboring communities, in which case the node will be assigned to the neighboring community with the highest occupancy rate.

## Initialization rules

Dynamic networks require us to perform initialization operations on the algorithm when dividing the network at each time step. The specific initialization operations are as follows.

### Initialize the weight vector

We use a uniformly distributed weight vector generation method (*Ji, Zhang & Zhou, 2020*) where the population size is denoted by $N$ and the $N$ weight vectors can be expressed as $\lambda^1, \ldots, \lambda^N$. Therefore, the population size and weight vectors can be set by expressing them as

$$N = C_{H+m-1}^{m-1} \tag{6}$$

where $N$ represents the population size and the number of weight vectors, $m$ represents the number of optimization objectives, and $H$ is the parameter to be set to solve for the desired number of populations by setting different parameters.

### Initialize neighborhood

The $T$ weight vectors closest to each weight vector are found by computing the Euclidean distance between any two weight vectors. The neighbors of each weight vector obtained are stored in $B(i)$ for all $i = 1$ to $N$, $B(i) = i_1, i_2, \ldots, i_T$.

### Initialize population

The Physarum-based network model PNM (*Gao et al., 2018b*) is used to initialize the population, PNM is an improvement on the mathematical model named PM (*Tero, Kobayashi & Nakagaki, 2007*), whose main improvement is the modification of single-entry single-exit to single-entry multiple-exit in the selection scheme for choosing whether the vertices are entrances or exits. The core mechanism of PNM is the feedback relationship between the cytoplasmic flux and the conductivity of the tube based on Poiseuille's law. The method can improve the efficiency of the community detection algorithm by identifying the inter-community edges from the intra-community edges in the network and further enhancing the positive feedback of the solving process in the algorithm. Therefore using this method for initializing populations can give a better structure of the network community at first.

### Initialize reference points

By calculating the fitness value of the initialized population, the maximum value of fitness of each target is used as the reference point $Z*$. In this article, we mainly use two target module degrees Q and NMI, so $Z*$ can be expressed as $Z* = \{NMI_{\max}, Q_{\max}\}$.

## Reorganization rules

The partitioning of dynamic network communities is solved in the decomposition-based MOEA/D framework, whose reorganization part uses a genomic matrix-based crossover and variation strategy combined with our proposed method of BNO assignment to find community solutions with high quality and high temporal smoothness.

### Crossover

To increase population diversity, we used a genome-based crossover strategy. To generate an individual, two different individuals from the neighbor population are selected as parents in a random way, and then different nodes are randomly selected to exchange the intra- and inter-community properties of the edges of the selected node connecting any node.

### Mutation

To avoid the population from falling into a local optimum, we use a genome-based mutation strategy. Individuals are selected with a certain probability whether they mutate or not, and individuals that mutate are selected in a random way to different nodes, changing the intra- and inter-community properties of the edges of the selected nodes connecting any node by taking the inverse operation.

### Boundary node occupancy allocation

To improve the local search ability of the community and the quality of the network community division, the nodes that are at the boundary should be attributed to the community whose neighbors have the highest percentage of the corresponding community, we propose the method of BNO assignment. Its details are made described in section 3.1.

## Update rules

In the dynamic network community partitioning process, we use the relevant update strategy in the decomposition-based multi-objective optimization algorithm in order to retain a better community structure. In network snapshots with time steps greater than 2, the Tchebycheff method is used to update the neighbors of each sub-problem as a way to eliminate the poorer community structure. For the neighbor $x^j (j \in B(i))$ of the i-th sub-problem $y_i'$ after reorganization, if $g^{te}(y_i'|\lambda^j, z) \leq g^{te}(x^j|\lambda^j, z)$, then the solution of that sub-problem will replace its neighbor. Then all solutions of $y_i'$ are added to the external population *EP*, and all dominated solutions are removed. Finally, the maximum modularity Q and NMI values are found and the reference point $Z* = new\{NMI_{max}, Q_{max}\}$ is updated.

## Rationalized selection strategy

The result we obtain in the partitioning process of dynamic network communities based on multi-objective optimization is a set of Pareto optimal solution sets consisting of multiple non-dominated solutions. We know that in the process of dynamic network community discovery, we need to choose a non-dominated solution from the optimal set of solutions at the current time step as the final pole at the current moment and also as a reference for the next moment. Most researchers choose the community structure with the highest snapshot quality, but this can easily lose the community structure with better time cost. Therefore, in order to accommodate such a situation, we propose to lose less snapshot quality in exchange for a better time cost. First we find the solution with the highest snapshot quality, then we use that solution as a reference to find other solutions in the range of 0.01 from it, and finally we select the solution with the best time cost from all the solutions in that range as the final solution.

**Table 1  The parameter setting for NOME in experiments.**

| Parameter | Value |
|---|---|
| Maxgen | 100 |
| pop | 100 |
| num_neighbor | 10 |
| p_mutation | 0.2 |
| num_repeat | 5 |

# EXPERIMENT

In this section, we conduct extensive experiments on synthetic and real datasets and analyze the results to evaluate our proposed algorithm. At the same time, we also compared the algorithm with classical algorithms in dynamic community discovery and advanced algorithms in recent years, including DYNMOGA, FacetNet (*Lin et al., 2009*), ECD, DECS (*Liu et al., 2020*), and ePMCL (*Wang et al., 2022*). The experimental results show that our proposed algorithm outperforms the comparison algorithm overall on the dynamic community discovery problem.

This experiment is running on Windows 10 Professional OS, 11th Gen Intel(R) Core(TM) i7-11700 @ 2.50 GHz and 16.0 GB RAM. The specific parameter settings of the algorithm are shown in Table 1: Maxgn is the maximum number of iterations, pop is the population size, num_neighbor is the number of neighbors, p_mutation is the mutation operator, and num_repeat is the number of times to run the algorithm. Due to the randomness of the algorithm process, we run the algorithm 5 times on each dataset to get the average result of each algorithm.

## Experiments on synthetic datasets

In this subsection, we conduct experiments on two synthetic datasets with known community divisions, use the normalized mutual information function NMI introduced in Section 2.2 to evaluate the accuracy of the algorithm, and analyze the experimental results.

### SYN-FIX datasets

The SYN-FIX datasets has 128 nodes in the network containing four communities, each community has 32 nodes at each time step, the average degree of each network is set to 16, and the edge connecting each node to a node that is not in the same community is *zout*. The size of *zout* determines the fuzziness of the network, and the larger the *zout* value the fuzzier the community structure. To simulate the evolutionary properties of dynamic networks, three nodes are randomly selected from each community to move to other communities at each time step, starting from the second time step. In this data, experiments were performed for two dynamic networks with *zout* = 3 and *zout* = 5, respectively, with a time step of 10.

As the results in Fig. 3 show, on the synthetic datasets SYN-FIX, algorithm NOME has the same NMI value of size 1 in Fig. 3A *zout* = 3 as algorithms ECD and ePMCL at all moments, both outperforming DYNMOGA and FacetNet. In Fig. 3B *zout* = 5, algorithms NOME and ePMCL have NMI values of 1 at all moments. The specific values for the five

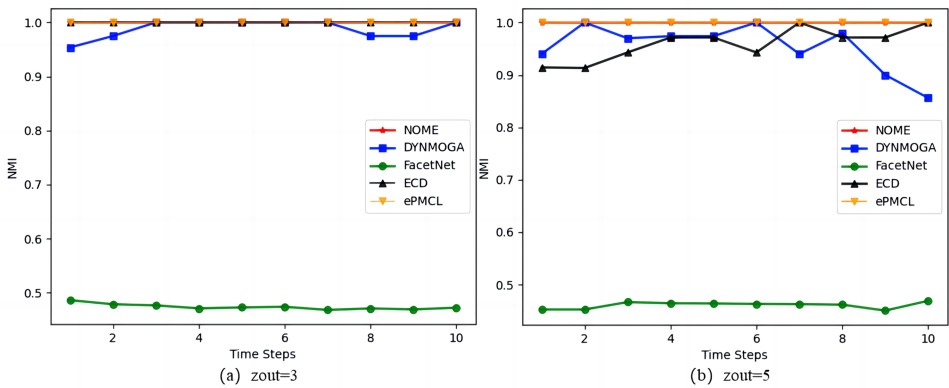

**Figure 3** Comparison of NMI values of NOME, DYNMOGA, FacetNet, ECD and ePMCL for synthetic datasets SYN-FIX on (A) *zout* = 3 and (B) *zout* = 5.

**Table 2** NMI values returned by NOME, DYNMOGA, FacetNet, ECD and ePMCL on the synthetic datasets SYN-FIX and *zout* =5. The best results are highlighted in bold.

| Time steps | 1 | 2 | 3 | 4 | 5 | 6 | 7 | 8 | 9 | 10 |
|---|---|---|---|---|---|---|---|---|---|---|
| NOME | **1.000** | **1.000** | **1.000** | **1.000** | **1.000** | **1.000** | **1.000** | **1.000** | **1.000** | **1.000** |
| DYNMOGA | 0.940 | **1.000** | 0.969 | 0.974 | 0.974 | **1.000** | 0.940 | 0.980 | 0.895 | 0.856 |
| FacetNet | 0.453 | 0.453 | 0.467 | 0.465 | 0.464 | 0.463 | 0.462 | 0.462 | 0.451 | 0.469 |
| ECD | 0.914 | 0.913 | 0.943 | 0.971 | 0.971 | 0.943 | **1.000** | 0.971 | 0.971 | **1.000** |
| ePMCL | **1.000** | **1.000** | **1.000** | **1.000** | **1.000** | **1.000** | **1.000** | **1.000** | **1.000** | **1.000** |

algorithms are given in Table 2, where DYNMOGA and ECD have NMI values of 1 for only two moments. Therefore NOME and ePMCL outperform other algorithms overall on this datasets.

### SYN-VAR datasets

The network of the SYN-VAR datasets consists of 256 nodes divided into four communities, each consisting of 64 nodes. Compared with the SYN-FIX datasets, SYN-VAR simulates the creation and disappearance of nodes, while the meaning of the parameter *zout* in SYN-VAR is the same as that of SYN-FIX. Experiments are also conducted in the SYN-VAR datasets for two dynamic networks with *zout* = 3 and *zout* = 5, respectively. In order to generate a dynamic network with a time step of 10, eight nodes are randomly selected from the four communities in each of the first five time steps to synthesize a new community consisting of 32 nodes as the newly generated nodes and communities in the next time step. In the subsequent five time steps, the nodes will return to their original communities, so the number of communities in this dynamic network with 10 time steps is 4, 5, 6, 7, 8, 8, 7, 6, 5 and 4. In addition, the average degree of each node in the community is set to half the community size, and at each time step, 16 nodes are randomly removed from the network, while 16 new nodes are added to the network.

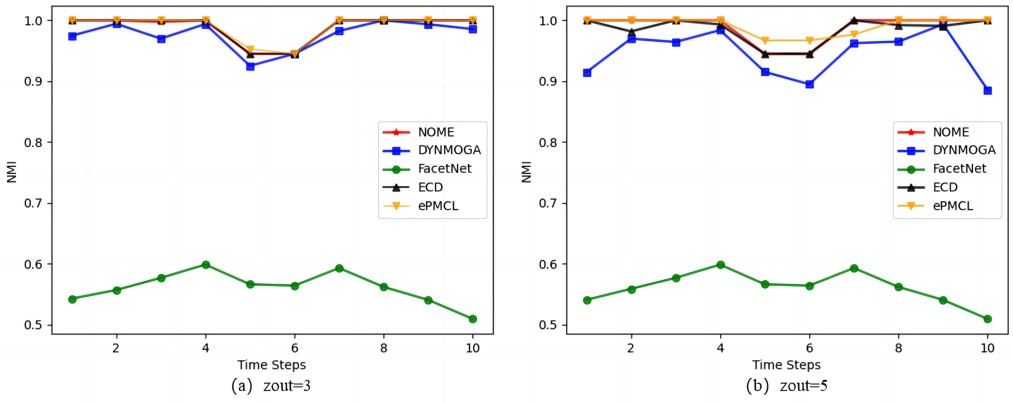

**Figure 4 Comparison of NMI values of NOME, DYNMOGA, FacetNet, ECD and ePMCL for synthetic datasets SYN-VAR on (A) *zout* = 3 and (B) *zout* = 5.**

Figure 4 shows the results of the five algorithms on the synthetic datasets SYN-VAR. In Fig. 4A *zout* = 3, algorithm ePMCL has slightly higher NMI values than NOME and ECD at moments 3 and 5, and all three are better than the other two algorithms overall. In Fig. 4B *zout* = 5, combined with Table 3, it is clear that NOME has the optimal NMI value for eight moments, ECD has the optimal NMI value for four moments, and ePMCL has the optimal NMI value for nine moments. Therefore, the ePMCL and NOME algorithms outperform the other algorithms overall in the synthetic datasets SYN-VAR.

## Experiments on real datasets

For the experiments in this subsection we use two real datasets, Cellphone Calls and Enron Mail, to evaluate our proposed algorithm using the modularity function of evaluation metrics and normalized mutual information NMI introduced in Section 2.2 and compare it with DECS, ePMCL and ECD.

### Cellphone calls datasets

The Cellphone Calls datasets, developed by VAST2008 Mini-Challenge3, consists of cellphone call records between members of the *Paraiso* movement, which cover 10 days of calls from 400 cellphones in June 2006. To build a dynamic network, in this datasets, a snapshot of the network at each time step consists of all phone call records during a day, where a node represents a phone and an edge between nodes represents a call between two phones that are available. Thus this data contains 10 network snapshots.

### Enron mail datasets

The Enron Mail datasets is derived from a collection of emails from 1999 to 2002 from a U.S. company called Enron Corporation. This experiment only selected email data from 2001 with potentially unusual emails, which contained 252,756 emails from 151 employees. It is divided by month into 12 copies representing 12 network snapshots, each snapshot consists of 151 nodes representing 151 employees. Also the node-to-node edges represent users with emails to and from each other.

**Table 3** NMI values returned by NOME, DYNMOGA, FacetNet, ECD and ePMCL on the synthetic datasets SYN-VAR and $zout = 5$. The best results are highlighted in bold.

| Time steps | 1 | 2 | 3 | 4 | 5 | 6 | 7 | 8 | 9 | 10 |
|---|---|---|---|---|---|---|---|---|---|---|
| NOME | **1.000** | **1.000** | **1.000** | **1.000** | 0.945 | 0.945 | **1.000** | **1.000** | **1.000** | **1.000** |
| DYNMOGA | 0.915 | 0.969 | 0.964 | 0.984 | 0.915 | 0.895 | 0.623 | 0.965 | 0.994 | 0.886 |
| FacetNet | 0.541 | 0.559 | 0.577 | 0.599 | 0.566 | 0.564 | 0.593 | 0.562 | 0.541 | 0.509 |
| ECD | **1.000** | 0.982 | **1.000** | 0.993 | 0.945 | 0.945 | **1.000** | 0.992 | 0.991 | **1.000** |
| ePMCL | **1.000** | **1.000** | **1.000** | **1.000** | **0.967** | **0.967** | 0.977 | **1.000** | **1.000** | **1.000** |

**Table 4** The mean and standard deviation of NMI and modularity returned by different algorithms on the real data set Cellphone Calls. The best result is highlighted in bold.

| Time steps | | 1 | 2 | 3 | 4 | 5 | 6 | 7 | 8 | 9 | 10 |
|---|---|---|---|---|---|---|---|---|---|---|---|
| NOME | NMI-mean | 0 | **0.643** | **0.716** | **0.714** | **0.718** | **0.728** | **0.715** | **0.731** | **0.744** | **0.712** |
| | NMI- ±std | 0 | **0.017** | **0.016** | 0.018 | 0.011 | **0.012** | **0.008** | **0.005** | **0.004** | **0.004** |
| | Q-mean | **0.636** | **0.648** | **0.630** | **0.636** | **0.645** | **0.640** | **0.633** | **0.615** | **0.625** | **0.623** |
| | Q- ±std | **0.002** | **0.006** | 0.007 | 0.006 | **0.006** | 0.006 | **0.004** | 0.006 | **0.002** | **0.004** |
| DECS | NMI-mean | 0 | 0.594 | 0.667 | 0.661 | 0.646 | 0.648 | 0.665 | 0.637 | 0.671 | 0.640 |
| | NMI- ±std | 0 | 0.019 | 0.018 | 0.017 | 0.016 | 0.044 | 0.037 | 0.015 | 0.011 | 0.021 |
| | Q-mean | 0.617 | 0.620 | 0.601 | 0.605 | 0.614 | 0.612 | 0.601 | 0.585 | 0.595 | 0.605 |
| | Q- ±std | 0.007 | 0.008 | **0.005** | 0.006 | 0.007 | **0.005** | 0.005 | 0.007 | 0.004 | **0.004** |
| ECD | NMI-mean | 0 | 0.590 | 0.648 | 0.650 | 0.633 | 0.643 | 0.653 | 0.626 | 0.663 | 0.609 |
| | NMI- ±std | 0 | 0.030 | 0.026 | **0.012** | **0.005** | 0.017 | 0.014 | 0.029 | 0.019 | 0.014 |
| | Q-mean | 0.616 | 0.621 | 0.596 | 0.598 | 0.612 | 0.613 | 0.607 | 0.586 | 0.597 | 0.607 |
| | Q- ±std | **0.002** | 0.011 | **0.005** | **0.001** | 0.007 | 0.007 | 0.006 | **0.003** | 0.013 | 0.007 |

Table 4 shows the mean and standard deviation of NMI and modularity returned by NOME, DECS, and ECD on the real dataset Cellphone Calls. It can be found that the mean value of NMI and the mean value of modularity obtained by our proposed algorithm during the evolution process on continuous time steps are much higher than other algorithms. In addition, the smaller the standard deviation, the more stable the algorithm. The standard deviation values of our proposed algorithm are mostly better than other algorithms, among which NMI accounts for seven out of 10-time scales and modularity accounts for 6. Therefore, our proposed method has better accuracy and stability than other community detection algorithms on this dataset.

Figure 5 shows the average NMI values obtained for NOME, DECS, ePMCL and ECD on Fig. 5A Cellphone Calls and Fig. 5B Enron Mail datasets. As we observed, our algorithm NOME has the highest NMI values at each time step on both real datasets, while ePMCL performs the worst. The gap between the remaining ECD and DECS algorithms is not very big, but there is still a certain gap compared with our proposed algorithm. The results show that NOME can significantly improve the similarity between the time steps of the community structure after using the strategy of rationalization rules, and also reveal that

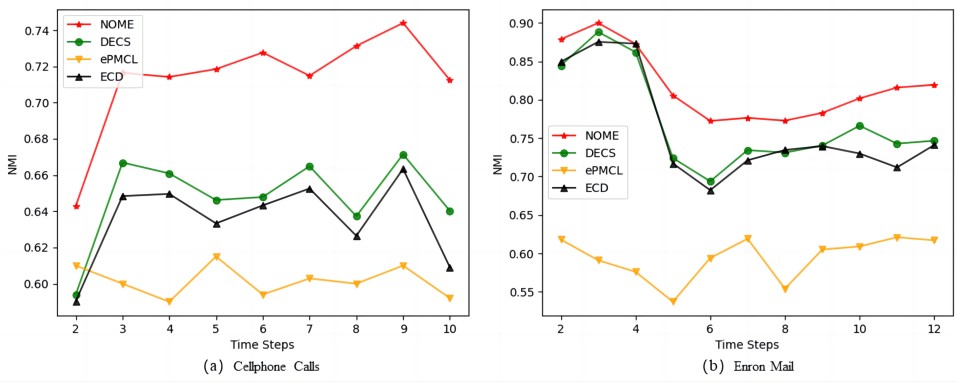

**Figure 5** Comparison of NMI values for NOME, DECS, ePMCL and ECD on real datasets (A) Cellphone Calls and (B) Enron Mail.

**Table 5** The mean, standard deviation, and difference of NMI returned by different algorithms on the real data set Enron Mail.

| Time steps | | 2 | 3 | 4 | 5 | 6 | 7 | 8 | 9 | 10 | 11 | 12 |
|---|---|---|---|---|---|---|---|---|---|---|---|---|
| NOME | mean | 0.879 | 0.900 | 0.873 | 0.806 | 0.772 | 0.776 | 0.773 | 0.783 | 0.802 | 0.816 | 0.819 |
| | ±std | 0.007 | 0.006 | 0.006 | 0.019 | 0.016 | 0.005 | 0.010 | 0.030 | 0.017 | 0.016 | 0.012 |
| DECS | mean | 0.845 | 0.888 | 0.862 | 0.724 | 0.694 | 0.734 | 0.731 | 0.740 | 0.766 | 0.743 | 0.747 |
| | ±std | 0.016 | 0.009 | 0.006 | 0.026 | 0.021 | 0.021 | 0.015 | 0.016 | 0.010 | 0.015 | 0.011 |
| ECD | mean | 0.849 | 0.875 | 0.873 | 0.717 | 0.682 | 0.721 | 0.735 | 0.740 | 0.730 | 0.712 | 0.741 |
| | ±std | 0.011 | 0.013 | 0.009 | 0.012 | 0.030 | 0.019 | 0.025 | 0.019 | 0.011 | 0.024 | 0.016 |
| DE-|NMI| | diff | 0.034 | 0.012 | 0.011 | 0.082 | 0.078 | 0.042 | 0.042 | 0.043 | 0.036 | 0.073 | 0.072 |
| EC-|NMI| | diff | 0.030 | 0.025 | 0.000 | 0.089 | 0.090 | 0.055 | 0.038 | 0.043 | 0.072 | 0.104 | 0.078 |

the time cost of the evolving community structure on the Cellphone Calls and Enron Mail datasets is better than other algorithms.

Tables 5 and 6, respectively, show the mean, standard deviation, and difference of NMI and the mean, standard deviation, and difference of modularity Q returned by the three algorithms NOME, DECS, and ECD on the real data set Enron Mail. Among them, DE-|NMI| represents the difference between NOME and DECS, EC-|NMI| represents the difference between NOME and ECD; DE-|Q| represents the difference between NOME and DECS, and EC-|Q| represents the difference between NOME and ECD. To further compare the pros and cons of the three different algorithms on the Enron dataset, we integrated the difference between their NMI and Q into Table 7 for comparison.

Figure 6 shows the modularity values of NOME, DECS and ECD on the real datasets. From Fig. 6A, we can see that the modularity values of the NOME algorithm are much higher than the other two algorithms at each moment on the Cellphone Calls datasets, so the performance of the NOME algorithm on the Cellphone Calls datasets is much better than the other algorithms. However, in Fig. 6B the NOME algorithm has slightly lower modularity values than the other two algorithms on the Enron Mail datasets for all moments except the first moment. To better evaluate the effectiveness of the algorithms,

**Table 6** The mean, standard deviation, and difference of the modularity returned by different algorithms on the real dataset Enron Mail.

| Time steps | | 1 | 2 | 3 | 4 | 5 | 6 | 7 | 8 | 9 | 10 | 11 | 12 |
|---|---|---|---|---|---|---|---|---|---|---|---|---|---|
| NOME | mean | 0.626 | 0.638 | 0.641 | 0.628 | 0.539 | 0.661 | 0.610 | 0.539 | 0.613 | 0.552 | 0.585 | 0.612 |
| | ±std | 0.003 | 0.019 | 0.004 | 0.010 | 0.011 | 0.013 | 0.010 | 0.010 | 0.014 | 0.013 | 0.006 | 0.003 |
| DECS | mean | 0.623 | 0.669 | 0.655 | 0.656 | 0.558 | 0.672 | 0.641 | 0.576 | 0.645 | 0.580 | 0.630 | 0.619 |
| | ±std | 0.022 | 0.006 | 0.003 | 0.012 | 0.009 | 0.005 | 0.006 | 0.002 | 0.008 | 0.005 | 0.006 | 0.004 |
| ECD | mean | 0.631 | 0.662 | 0.662 | 0.657 | 0.551 | 0.673 | 0.639 | 0.574 | 0.620 | 0.556 | 0.618 | 0.624 |
| | ±std | 0.003 | 0.012 | 0.004 | 0.005 | 0.020 | 0.012 | 0.011 | 0.008 | 0.016 | 0.030 | 0.008 | 0.003 |
| DE-|Q| | diff | 0.003 | 0.031 | 0.014 | 0.028 | 0.019 | 0.011 | 0.031 | 0.037 | 0.032 | 0.028 | 0.045 | 0.007 |
| EC-|Q| | diff | 0.005 | 0.024 | 0.021 | 0.029 | 0.012 | 0.012 | 0.029 | 0.035 | 0.007 | 0.004 | 0.033 | 0.012 |

**Table 7** For the comparison of the difference between |NMI| and |Q| in Tables 5, 6, the best results are highlighted in bold.

| Time steps | 1 | 2 | 3 | 4 | 5 | 6 | 7 | 8 | 9 | 10 | 11 | 12 |
|---|---|---|---|---|---|---|---|---|---|---|---|---|
| DE-|NMI| | 0.000 | **0.034** | 0.012 | 0.011 | **0.082** | **0.078** | **0.042** | **0.042** | **0.043** | **0.036** | **0.073** | **0.072** |
| DE-|Q| | 0.003 | 0.031 | **0.014** | **0.028** | 0.019 | 0.011 | 0.031 | 0.037 | 0.032 | 0.028 | 0.045 | 0.007 |
| EC-|NMI| | 0.000 | **0.030** | **0.025** | 0.000 | **0.089** | **0.090** | **0.055** | **0.038** | **0.043** | **0.072** | **0.104** | **0.078** |
| EC-|Q| | 0.005 | 0.024 | 0.021 | **0.029** | 0.012 | 0.012 | 0.029 | 0.035 | 0.007 | 0.004 | 0.033 | 0.012 |

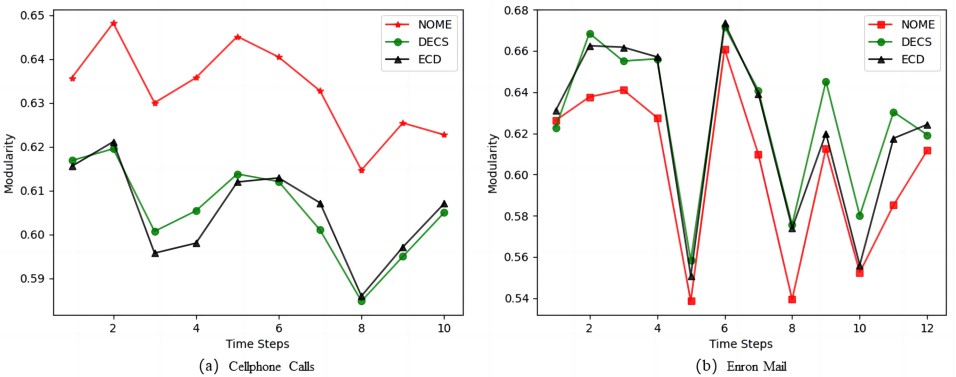

**Figure 6** Comparison of Modularity values of NOME, DECS and ECD on real datasets (A) Cellphone Calls and (B) Enron Mail.

we compare the three algorithms on the Enron Mail datasets in terms of the difference between NMI and modularity, and we expect a smaller modularity in exchange for a larger boost in NMI on this datasets.

From Table 7, we can see that in the comparison between NOME and DECS, there are nine best results for NMI and only two best results for modularity, and in the comparison between NOME and ECD, there are 10 best results for NMI and only one best result for modularity. Therefore, the results show that the strategy of border node occupancy distribution proposed by us can better assign nodes to the correct community, to improve the quality of community division. Although affected by the rationalized selection strategy, some sacrifices may have better community results, but overall it can be concluded that

the NOME algorithm has better accuracy in dynamic community discovery than other advanced algorithms.

## CONCLUSIONS AND FUTURE RESEARCH

This study proposes a new model to improve the dynamic community discovery problem in complex networks using the NOME algorithm. The model adopts a decomposition-based multi-objective evolutionary clustering framework, takes modularity and normalized mutual information as optimization objectives, and optimizes the time cost and snapshot quality of dynamic network community discovery. To maximize the snapshot quality, improve the clustering accuracy, and make the divided network have the relationship that the nodes in the community are tightly connected and the nodes in the community are sparsely connected, we propose a strategy of boundary node occupancy. It can grasp the relationship between nodes very well, ensure that nodes can be divided into better communities, and thus improve the clustering accuracy of communities. In addition, we propose a rationalized selection strategy to minimize the cluster drift between two consecutive time steps, so that the network can obtain the community with greater similarity to the previous community when selecting the final solution of the community partition. Finally, the proposed algorithm is evaluated by comparative experiments on synthetic datasets and real datasets. Compared with classic and recent popular algorithms, it is proved that the proposed algorithm has better performance than other algorithms in dynamic network community discovery. In the process of network community division, we did not consider that there may be some relationship between nodes, and the same node may have problems in different community structures. Therefore, the algorithm is only suitable for unweighted and undirected network models. In future research work, we consider trying to develop overlapping or empowered social network discovery.

### Funding

This work was supported by the National Natural Science Foundation of China under Grant Nos (61862051, 62241206); the Science and Technology Foundation of Guizhou Province under Grant Nos (ZK[2022]549,ZK[2022]520) and the Educational Department of Guizhou under Grant Nos ([2019]203, KY[2019]067). The funders had no role in study design, data collection and analysis, decision to publish, or preparation of the manuscript.

### Grant Disclosures

The following grant information was disclosed by the authors:
The National Natural Science Foundation of China: 61862051, 62241206.
The Science and Technology Foundation of Guizhou Province: ZK[2022]549, ZK[2022]520.
The Educational Department of Guizhou: [2019]203, KY[2019]067.

## Competing Interests

The authors declare there are no competing interests.

## Author Contributions

- Lei Cai conceived and designed the experiments, performed the experiments, analyzed the data, performed the computation work, prepared figures and/or tables, authored or reviewed drafts of the article, and approved the final draft.
- Jincheng Zhou conceived and designed the experiments, performed the experiments, performed the computation work, prepared figures and/or tables, authored or reviewed drafts of the article, and approved the final draft.
- Dan Wang conceived and designed the experiments, analyzed the data, performed the computation work, authored or reviewed drafts of the article, and approved the final draft.

## Data Availability

The data is available at the HCIL Archive and the Enron Email Dataset:

- Available at http://www.cs.umd.edu/hcil/VASTchallenge08/
- Available at http://www.cs.cmu.edu/~enron/

The raw data and code are available in the Supplementary Files.

## Supplemental Information

Supplemental information for this article can be found online at http://dx.doi.org/10.7717/peerj-cs.1477#supplemental-information.

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
