# Peer review of "Improving temporal smoothness and snapshot quality in dynamic network community discovery using NOME algorithm"

_PeerJ Computer Science, doi:10.7717/peerj-cs.1477_

## Round 0.1 · original submission · Major Revisions

Dear authors,

The reviews for your manuscript are included at the bottom of this letter. We ask that you make changes to your manuscript based on those comments.

Best wishes,

Reviewer 1 ·

Basic reporting

1-The English used in the study is written in an understandable way.
2-While summarizing the relevant studies, their contributions to science should be examined with their deficiencies or weaknesses. "Evolutionary clustering refers to the evolution of community structure over time (Sun et al.,2022)." I do not find such a quotation useful. All attributions should be checked and any references in the situation I described should be restated.

Experimental design

3- The motivation of the study and its contributions to the world of science should be explained in a more understandable way.

Validity of the findings

4- Express the limitations of the study, if any.

Additional comments

The authors proposed a dynamic community discovery algorithm in this study. They stated that thanks to the proposed algorithm, they reduced the time cost with some loss in snapshot quality. The study is very interesting and I think that it will contribute to the related field.

Reviewer 2 ·

Basic reporting

• Keywords should be listed alphabetically.
• In the abstract, the quantitative results should be properly specified and reported.
• The abstract should be clearer to the reader about the proposed method and its justification. The abstract should summarize the primary contributions, the proposed technique, the main problem, the acquired results, the benchmark tests, the comparison methodologies, and so on.
• Authors should explicitly state the motivation for their paper. It is unclear why specifically a new multi-objective evolutionary algorithm (NOME) is utilized for the focused community discovery problem.
• The authors should go over the literature review in greater depth and clarity. Make the paragraphs in the introductory section more complete; they are relatively brief. The current introduction appears to be straightforward and lacks some substance relating to problem formulation.
• The authors just mentioned the researchers' relevant works, but they did not assess the benefits and drawbacks of those works. Please assess how their research differs from others in the related work area. What do they possess that others do not? Why or how are they superior? The current introduction appears to be straightforward and lacks some substance relating to problem formulation. The authors should concentrate on the main topic of the study and give a Literature Review in the form of tables to make research gaps and innovations visible. There is no authoritative synthesis assessing the current state of the art.
• Recent literature is not deeply explored. The listed references seem a bit old for this hot research topic. Consequently, the real novelty of the paper is not highlighted.
• Authors should include a paragraph in the introduction. To highlight the major works, they should write "The main contributions of this paper are: (i)..... (ii)....... and (iii)......." Authors should present an additional motivating factor and explain the paper's originality in this manner.
• “Concept and Related Work” section does not involve the related works on the focused problem.
• Some mathematical notations are not rigorous enough to properly understand the paper's contents. The authors are asked to double-check all variable definitions and further clarify these equations. All variables and their intervals should be defined.
• Some equations should be accompanied by proper citations. They appear to be proposed and utilized for the first time in this paper.
• Equations must be utilized with the proper numbers. All equation numbers must be double-checked.
• All variables in the equations should be stated in italics.
• Some paragraphs are difficult to read. They should be broken into two or more sections for readability and comprehension.

Experimental design

• More metrics may be used for comparison of simulation results. For example, more quality metrics for multi-objective optimization problems may be used for experimental results. It will be better to show the quality of the obtained Pareto solution set.

Validity of the findings

• Findings should be presented in a systematic manner and should respond to the goal of the study. Discuss your findings in terms of what was previously known and what was not previously known regarding the topic of your research.
• The study lacks an analysis of the outcomes. There is a disconnect between the results and the conclusion. There should be a comparison of the outcomes of these two sections. The authors must be able to interpret the results and relate them to the structure of the algorithms after comparing the approaches. It would be interesting to hear the writers' ideas on why the method works the way it does. Such analyses would be at the heart of the study, demonstrating a comprehension of the reasons behind the outcomes.
• Clarifying the study's limitations assists readers to better understand how the results should be interpreted under what conditions. A good assessment of a study's limits demonstrates that the researcher has a comprehensive understanding of his or her work. However, the authors' paper fails to illustrate this. The authors should explain the advantages and disadvantages of their methodology. What are the work's limitations and methodology(ies)? Please describe the practical benefits and address the research limitations. The pros and weaknesses of the proposed algorithm must be discussed in the results discussion. These constraints can be arranged around simple distinctions of who, what, where, when, why, and how you chose in your research. There is a need for an unbiased viewpoint in the paper. Can this method be generalized for dynamic overlapping networks?
• There is no statistical test to assess the method's outcomes' significance. The conclusion cannot be maintained without such a statistical test. The evolutionary technique utilized in the research is stochastic in nature, which means that different outcomes may be obtained in various runs. As a result, standard deviations should be provided. Statistical test results may be included as well.
• Considering the experimental results, some further recommendations and conclusions should be discussed in the paper. The conclusion is inadequate. Furthermore, there is no space for discussing the results. The conclusion part requires extensive rewriting. It should briefly discuss the study's findings as well as potential additional research directions. In the conclusion section, the writers should discuss academic implications, important discoveries, weaknesses, and future research directions. In general, the conclusion in its current form is perplexing. Concerning the Conclusion section, it would be better to call it "Conclusions and Future Research," and it is strongly urged that this manuscript's future research be included. What is going to happen next? What should we expect from future papers? Rewrite it with the following suggestions in mind: - Highlight your analysis and just include the most significant elements for the entire document. - Mention the advantages. - At the end of this section, mention the implication that was drawn.

Additional comments

In this paper, the authors present a dynamic community discovery approach based on multi-objective evolutionary clustering and node occupancy assignment that aims to improve dynamic community discovery snapshot quality and time cost. The research appears to be original, and intriguing findings are provided. Below is a list of the reviewer's recommendations, queries, and issues with the paper:


• Keywords should be listed alphabetically.
• In the abstract, the quantitative results should be properly specified and reported.
• The abstract should be clearer to the reader about the proposed method and its justification. The abstract should summarize the primary contributions, the proposed technique, the main problem, the acquired results, the benchmark tests, the comparison methodologies, and so on.
• Authors should explicitly state the motivation for their paper. It is unclear why specifically a new multi-objective evolutionary algorithm (NOME) is utilized for the focused community discovery problem.
• The authors should go over the literature review in greater depth and clarity. Make the paragraphs in the introductory section more complete; they are relatively brief. The current introduction appears to be straightforward and lacks some substance relating to problem formulation.
• The authors just mentioned the researchers' relevant works, but they did not assess the benefits and drawbacks of those works. Please assess how their research differs from others in the related work area. What do they possess that others do not? Why or how are they superior? The current introduction appears to be straightforward and lacks some substance relating to problem formulation. The authors should concentrate on the main topic of the study and give a Literature Review in the form of tables to make research gaps and innovations visible. There is no authoritative synthesis assessing the current state of the art.
• Recent literature is not deeply explored. The listed references seem a bit old for this hot research topic. Consequently, the real novelty of the paper is not highlighted.
• Authors should include a paragraph in the introduction. To highlight the major works, they should write "The main contributions of this paper are: (i)..... (ii)....... and (iii)......." Authors should present an additional motivating factor and explain the paper's originality in this manner.
• Findings should be presented in a systematic manner and should respond to the goal of the study. Discuss your findings in terms of what was previously known and what was not previously known regarding the topic of your research.
• More metrics may be used for comparison of simulation results. For example, more quality metrics for multi-objective optimization problems may be used for experimental results. It will be better to show the quality of the obtained Pareto solution set.
• “Concept and Related Work” section does not involve the related works on the focused problem.
• Some mathematical notations are not rigorous enough to properly understand the paper's contents. The authors are asked to double-check all variable definitions and further clarify these equations. All variables and their intervals should be defined.
• Some equations should be accompanied by proper citations. They appear to be proposed and utilized for the first time in this paper.
• Equations must be utilized with the proper numbers. All equation numbers must be double-checked.
• All variables in the equations should be stated in italics.
• The study lacks an analysis of the outcomes. There is a disconnect between the results and the conclusion. There should be a comparison of the outcomes of these two sections. The authors must be able to interpret the results and relate them to the structure of the algorithms after comparing the approaches. It would be interesting to hear the writers' ideas on why the method works the way it does. Such analyses would be at the heart of the study, demonstrating a comprehension of the reasons behind the outcomes.
• Clarifying the study's limitations assists readers to better understand how the results should be interpreted under what conditions. A good assessment of a study's limits demonstrates that the researcher has a comprehensive understanding of his or her work. However, the authors' paper fails to illustrate this. The authors should explain the advantages and disadvantages of their methodology. What are the work's limitations and methodology(ies)? Please describe the practical benefits and address the research limitations. The pros and weaknesses of the proposed algorithm must be discussed in the results discussion. These constraints can be arranged around simple distinctions of who, what, where, when, why, and how you chose in your research. There is a need for an unbiased viewpoint in the paper. Can this method be generalized for dynamic overlapping networks?
• Some paragraphs are difficult to read. They should be broken into two or more sections for readability and comprehension.
• There is no statistical test to assess the method's outcomes' significance. The conclusion cannot be maintained without such a statistical test. The evolutionary technique utilized in the research is stochastic in nature, which means that different outcomes may be obtained in various runs. As a result, standard deviations should be provided. Statistical test results may be included as well.
• Considering the experimental results, some further recommendations and conclusions should be discussed in the paper. The conclusion is inadequate. Furthermore, there is no space for discussing the results. The conclusion part requires extensive rewriting. It should briefly discuss the study's findings as well as potential additional research directions. In the conclusion section, the writers should discuss academic implications, important discoveries, weaknesses, and future research directions. In general, the conclusion in its current form is perplexing. Concerning the Conclusion section, it would be better to call it "Conclusions and Future Research," and it is strongly urged that this manuscript's future research be included. What is going to happen next? What should we expect from future papers? Rewrite it with the following suggestions in mind: - Highlight your analysis and just include the most significant elements for the entire document. - Mention the advantages. - At the end of this section, mention the implication that was drawn.

---

## Round 0.2 · accepted · Accept

Dear authors,

Thank you for clearly addressing the reviewers' comments. Your article is accepted for publication now.

Best wishes,

Reviewer 1 ·

Basic reporting

It is seen that the author has eliminated all the deficiencies that I have stated in the revision.

Experimental design

It is seen that the author has eliminated all the deficiencies that I have stated in the revision.

Validity of the findings

It is seen that the author has eliminated all the deficiencies that I have stated in the revision.

Additional comments

It is seen that the author has eliminated all the deficiencies that I have stated in the revision.